# Exploring Asthma as a Protective Factor in COVID-19 Outcomes

**DOI:** 10.3390/ijms26041678

**Published:** 2025-02-16

**Authors:** Anthony E. Quinn, Lei Zhao, Scott D. Bell, Muhammad H. Huq, Yujiang Fang

**Affiliations:** 1Department of Microbiology, Immunology & Pathology, College of Osteopathic Medicine, Des Moines University, West Des Moines, IA 50266, USA; anthony.e.quinn@dmu.edu (A.E.Q.); scott.d.bell@dmu.edu (S.D.B.); muhammad.h.huq@dmu.edu (M.H.H.); 2The Department of Respiratory Medicine, the 2nd People’s Hospital of Hefei and Hefei Hospital Affiliated to Anhui Medical University, Hefei 230002, China; 19955873766@163.com; 3Department of Surgery, University of Missouri School of Medicine, Columbia, MO 65212, USA; 4Ellis Fischel Cancer Center, University of Missouri School of Medicine, Columbia, MO 65212, USA

**Keywords:** asthma, COVID-19, allergy

## Abstract

Asthma has long been associated with increased susceptibility to viral respiratory infections, leading to significant exacerbations and poorer clinical outcomes. Contrarily and interestingly, emerging data and research surrounding the COVID-19 pandemic have shown that patients with asthma infected with SARS-CoV-2 experienced decreased severity of disease, lower hospitalization rates, as well as decreased morbidity and mortality. Research has shown that eosinophils could enhance immune defense against viral infections, while inhaled corticosteroids can assist in controlling systematic inflammation. Moreover, reduced ACE-2 expression in individuals with asthma may restrict viral entry, and the Th2 immune response may offset the Th1 response typically observed in severe COVID-19 patients. These factors may help explain the favorable outcomes seen in asthmatic patients during the COVID-19 pandemic. This review highlights potential protective mechanisms seen in asthmatic patients, including eosinophilia, the use of inhaled corticosteroids, reduced ACE-2 expression, and a dominate Th2 immune response. Such a study will be helpful to better manage patients with asthma who have contracted COVID-19.

## 1. Introduction

Asthma is a common chronic inflammatory airway disease characterized by reversable airway obstruction that presents with symptoms of wheezing, shortness of breath, and coughing [1]. Asthma is typically driven by the induction and activation of inflammatory cells in the airway such as eosinophils, mast cells, and T cells. The cells can release inflammatory mediators that cause an increase in bronchoconstriction, and thus closure of the airways. During an asthma exacerbation, further obstruction of the airway occurs as airway smooth-muscle contraction and mucus hypersecretion increases. The most common triggers for an asthma attack are allergens, air pollution, and even the weather.

Worldwide, asthma affects around 260 million individuals [2]. In the United States, the CDC estimates asthma affects about 25 million Americans (7.7%), with higher prevalence in male children compared to female children, 7.3% and 5.6%, respectively, which then switches to a higher prevalence in adult females than adult males, 9.7% and 6.2%, respectively. By ethnicity, American Indians and Alaska Natives show a notably higher prevalence at 12.3%, followed by Black people at 10.9% as opposed to 7.6% for White people. Additionally, Asian and Hispanic individuals show a notably lower prevalence, at 4.1% and 5.4%, respectively. Socioeconomic status also shows a trend with asthma rates, at 10.4% for people living below 100% of the poverty threshold in contrast to 6.8% for people living 450% or higher above the poverty threshold [3].

Asthma has historically been shown to predispose patients to viral respiratory infections and the subsequent exacerbation of symptoms, leading to poorer prognosis [4]. With the recent Coronavirus Disease 2019 (COVID-19) pandemic, concerns arose regarding the potential exacerbation of symptoms for asthma patients infected with the severe acute respiratory syndrome coronavirus 2 (SARS-CoV-2). Emerging research has shown that contrary to other viral respiratory infections, asthma patients infected with SARS-CoV-2 during the COVID-19 pandemic had a lower risk of hospitalization, and no increased morbidity or mortality [5].

Asthma is a chronic inflammatory respiratory disorder characterized by airway inflammation leading to obstructed airflow and bronchial hyperresponsiveness, leading to symptoms including wheezing, coughing, and dyspnea [6]. Typically, asthma symptoms are exacerbated by triggers such as allergens or infections [7].

SARS-CoV-2 is a virus transmitted through respiratory droplets, with symptom presentation ranging from asymptomatic to multi-organ injury [8]. Clinically, the disease presents as a flu-like illness with cough, sore throat, fever, headache, fatigue, and breathlessness [9]. During the height of the pandemic, there were an estimated 460,000 deaths in 2021 alone [10]. Additionally, a reported 10% of patients still experience long-term effects including neuropsychiatric, neurological, cardiovascular, and hematological problems post SAR-CoV-2 diagnosis [11].

While severe cases of SARS-CoV-2 unproportionally affected individuals with comorbidities, representing 61.7% of all cases [12], asthma patients followed an unexpected trend. While the underlying reasons remain unknown, there have been a variety of studies investigating potentially protective factors predisposing asthma patients to more favorable outcomes. These include asthma-induced eosinophilia, a T-helper type 2 cell (Th2)-skewed immune response, angiotensin-converting enzyme 2 (ACE-2), and inhaled corticosteroids (ICS).

Emerging research has suggested that eosinophilia, a common feature of asthma, may play a protective role in asthmatic patients infected with SARS-CoV-2. Eosinophils, known for their involvement in allergic responses, can act as immune mediators via multiple mechanisms to promote antigen presentation and facilitate T cell responses. Studies have shown that asthmatic patients with higher eosinophil counts may act as a protective factor against COVID-19, preventing disease progression and severity [13].

Atopic asthma is characterized by a Th2-skewed immune response, marked by the activation of Th2 cells that release cytokines such as Interleukin 4 (IL-4), IL-5, and IL-13, which can lead to eosinophilia, mucous production, respiratory inflammation, and airway remodeling [14]. The Th2 response rivals the Th1 immune system activation typically seen in viral infections like SARS-CoV-2 [15]. The Th2 cytokines seen in asthma, especially IL-4 and IL-13, may downregulate the Th1 immune response, potentially offering a protective mechanism against COVID-19 and could provide an explanation as to why asthmatic patients infected with SARS-CoV-2 have better disease outcomes.

ACE-2, crucial for SARS-CoV-2’s entry into cells, is expressed in multiple organs, including the respiratory tract [16]. In asthmatic individuals, especially with atopic asthma, ACE-2 expression is generally lower in airway cells, potentially serving as a protective factor against severe COVID-19 [17]. The reduced ACE-2 expression in patients with asthma may contribute to a lower risk of poor COVID-19 outcomes.

ICSs, the first-line treatment for asthma, may offer additional protection against severe COVID-19 outcomes. They reduce airway inflammation and modulate immune responses by downregulating pro-inflammatory cytokines, many of which are elevated during SARS-CoV-2 infection. Their ability to mitigate cytokine storms, a hallmark of severe COVID-19, may support that patients on inhaled corticosteroid therapy experienced favorable clinical outcomes [18].

## 2. Eosinophilia

Elevated blood eosinophil counts, termed eosinophilia, is a hallmark of allergic diseases [19], and has a marked presence in up to 50% of asthma cases [20]. Differentiated and matured in bone marrow, eosinophils migrate to and reside in tissues, most commonly those of the respiratory and gastrointestinal tract [21]. Eosinophilia in allergic diseases arises from a complex cascade of immune responses that lead to eosinophil recruitment, activation, and increased persistence in affected tissues. Eosinophilopoiesis begins with the presentation of an allergen by antigen-presenting cells (APCs), particularly dendritic cells to Th2 lymphocytes [22]. Once sensitized to the allergen, Th2 cells produce and release cytokines like IL-4, IL-13, and IL-5 [19]. IL-5 has additional crucial roles in this response, as it stimulates additional eosinophil production in bone marrow, and prolongs eosinophil survival by inhibiting pro-apoptotic factors [23,24]. Concurrently, chemokines such as eotaxins, which are produced by epithelial and immune cells, bind to receptors on eosinophils, directing their migration to inflamed sites [25]. Once at the target tissues, eosinophils release granule proteins, including major basic proteins (MBPs) and eosinophil cationic proteins (ECPs), causing tissue damage and inflammation [26].

Eosinophilic asthma arises from a variety of genetic and environmental factors including exposure to pollen, pollutants, respiratory infections, or physiological factors like obesity [27]. A combination of these factors can trigger the Th2 immune response in the airway, leading to chronic inflammation, and eventually resulting in airway remodeling [28]. In the absence of such stimuli, eosinophils undergo apoptosis and are cleared by macrophages, preventing inflammation [29]. Typically, eosinophils circulate for 8 to 18 h and remain in tissues for 3 to 4 days, and are programmed to die without viability-promoting factors [30]. However, the mechanisms of cytokines like IL-3, IL-5, IL-9, IL-13, IL-15, and granulocyte–macrophage colony-stimulating factor (GM-CSF), which are abundant in asthmatic lungs, prolong eosinophil survival by inhibiting apoptosis [31]. Compounding on this response, cytokines released by airway epithelial cells, thymic stromal protein (TSLP), IL-25, and IL-33, have been shown to act together to not only illicit a Th2 immune response, leading to airway modifications, but also act on eosinophil function by the inhibition of apoptosis as well [30]. Furthermore, eosinophil adhesion to extracellular matrix proteins, such as fibronectin, triggers the autocrine release of GM-CSF, IL-3, and IL-5, prolonging their lifespan for weeks and contributing to chronic inflammation [30]. The mechanisms of cytokines and extracellular matrix proteins not only allow for an increased longevity, but for the abundant accumulation of eosinophils seen in asthma.

While they are mostly correlated with allergic disease and its exacerbation, more recent research has posed that eosinophils possess mechanisms that aid in battling viral diseases. (Figure 1). Eosinophils create and store ribonuclease (RNase) proteins, specifically eosinophil-derived neurotoxin (EDN) and ECP, which can degrade viral RNA [32], interfering with the virus’s ability to replicate. RNases are cytotoxic proteins that contribute to inflammation and remodeling in allergic and asthmatic disease states but can conversely function as anti-viral and immune-modulating proteins in the presence of a viral infection. Furthermore, eosinophils can also produce nitric oxide (NO), which has the ability to inhibit viral replication by multiple mechanisms via the enzyme NO synthase [33]. Both RNases and NO can further enhance immune responses by increasing antigen presentation and T cell activation. RNases help degrade viral RNA, reducing the viral load, allowing APCs to process and display viral antigens more effectively, facilitating recognition by T cells [34]. NO does so by upregulating T cell proliferation [35]. Research has shown that eosinophils can also express viral sensing receptors, acting as APC-like molecules. Emerging evidence has shown that eosinophils can present retinoic acid-inducible gene I (RIG I) on their surface, recognizing viral RNA sequences, and triggering an immune response [36,37,38]. Eosinophils have also been shown to express major histocompatibility complex class II (MHC-II) molecules on their surface, which bind to antigenic peptides from extracellular proteins processed by antigen-presenting cells, and stimulate a T cell response [34,39].

The innate immunoprotective functions of eosinophils could provide a protective response against viruses, like SARS-CoV-2, especially in patients with elevated counts seen in asthma and other allergic diseases. Although asthma has typically been correlated with more severe viral respiratory infections [40], research collected throughout the pandemic showed no association between asthma and severe COVID-19 illness [41]. Increasing evidence has shown that eosinophilia seen in patients with asthma may lower COVID-19 disease severity.

In a retrospective cohort study, Liu et al. compared mortality and sequential organ failure assessment (SOFA) scores to compare COVID-19 outcomes in patients with asthma, COPD, and no airway disease. The median SOFA score and mortality for patients with no airway disease was 0.32 and 11%, respectively [42]. In comparison, patients with asthma had a lower median SOFA score of 0.15 and lower mortality after adjusting for confounding variables [42]. High eosinophil levels, ≥200 cells/μL, directly correlated with the lower mortality in all groups. However, asthma patients still showed significantly improved outcomes after adjusting for eosinophilia, suggesting even non-eosinophilic asthmatics still benefit from protection [42].

A multicenter retrospective study conducted on outpatients and inpatients presenting to six hospitals in the Mount Sinai Health System, New York between March and June 2020 analyzed patients diagnosed with COVID-19, with and without asthma. The study found that patients with asthma had significantly lower mortality (OR = 0.64), hospitalization rates (OR = 0.43) and admission to the intensive care unit (OR = 0.43) [43]. Similarly to Liu et al., Ho et al. found that patients with an eosinophil count ≥200 cells/μL with or without asthma had a 91% and 84% lower morality, respectively, than individuals with an eosinophil count <200 cells/μL [43]. In comparison to allergic asthma dominated by eosinophils, non-allergic asthma patients showed a 48% increase in severe COVID-19 infections [43].

Ferastraoaru et al. examined the outcomes of asthmatic patients with COVID-19 and analyzed the relationship between asthma, eosinophil counts, and disease outcomes. The study found that patients with absolute eosinophil counts (AECs) ≥ 150 cells/μL were less likely to be admitted to the hospital [44]. Furthermore, among hospitalized asthmatics, those with peak AECs ≥150 cells/μL had a lower mortality rate (9.6%) compared to those with lower AECs (25.8%) [44]. They also found that mortality rates in patients with asthma alone, without any other comorbidities like heart failure, chronic kidney disease, and chronic obstructive pulmonary disease, or hypertension, were similar to that of patients without asthma or any comorbidities [44].

These studies collectively demonstrate that asthma, especially when associated with higher eosinophil counts, might have a protective effect against severe outcomes in COVID-19. Liu et al., Ho et al., and Ferastraoaru et al. all reported that asthmatic patients, especially those with eosinophil levels ≥ 150 cells/μL, experienced lower mortality and better clinical outcomes compared to non-asthmatic patients [42,43,44]. These findings underscore the relationship between eosinophil level and COVID-19 outcomes and indicate that asthma may not increase risk for severe COVID-19, contrary to initial assumptions, but may provide a greater immune defense against the virus.

## 3. Th2-Cell-Skewed Immune Response

Type 2 immune responses are marked by elevated type 2 cytokines and increased blood eosinophil counts [45]. Allergic diseases, such as atopic asthma, prompt the activation of Th2 cells, which are a crucial component of the immune response. This activation of these cells results in the release of several key type 2 cytokines, IL-4, IL-5, and IL-13 [46]. These cytokines play distinct and important roles in the progression of allergic inflammation.

IL-4 is responsible for the differentiation of B lymphocytes into plasma cells [47], naïve CD4+ T cells into Th2 cells [48], and immunoglobulin (Ig) class switching to IgG1 and IgE production in B cells [45,49]. IL-13 is an important cytokine that promotes mucus production, tissue remodeling, and airway hyperreactivity [50]. Both IL-4 and IL-13 play a critical role in suppressing the type 1 immune response, which is predominately seen during viral infections, limiting excessive inflammation by inhibiting Th1 cell production (Figure 2) [45,51]. IL-4 inhibits Th1 differentiation by downregulating the expression of the IL-12R β2 (Interleukin 12 receptor, beta 2 subunit), a critical component of IL-12 signaling that drives Th1 cell development. Without IL-12R β2, T cells cannot respond to IL-12, effectively preventing Th1 differentiation by inhibiting the differentiation of naïve CD4+ T cells into Th1 or Th17 effector cells [52,53]. IL-4 and IL-13 also activates the signal transducer and activator of transcription 6 (STAT6), which acts as a Th2-inducing transcriptional activator [54]. STAT6 binds to the interferon gamma (IFNγ)-responsive element (γRE) motif, inhibiting IFNγ, a cytokine crucial to Th1 immune responses [55]. IL-5′s role in Th2 immune responses involves driving the activation and maturation of eosinophils, which plays a central role in allergic inflammation [56]. Elevated IL-5 levels lead to an increased accumulation of eosinophil in the lungs, contributing to airway hyperreactivity and chronic inflammation [57]. However, IL-5 may provide a defense mechanism in certain inflammatory conditions, potentially affecting the immunologic effects in viral infections by promoting tissue repair and modulating immune-driven damage [58].

The viral immunopathology of SARS-CoV-2 involves the activation of Th1 cells, which release cytokines such as IL-1 and IFNγ [59]. IL-1 is a potent pro-inflammatory cytokine that plays a central role in the immune response, particularly by activating immune cells and promoting inflammation [60]. IFNγ along with the activation of CD40, allows for the activation of macrophages, leading to the phagocytosis of the infected host cells [61]. Furthermore, macrophages increase their pro-inflammatory cytokine production and enhance their phagocytic and cytolytic capabilities [62]. This Th1-dominant response can lead to a chronic inflammatory reaction that can lead to severe symptoms for the infected patient. However, the same response allows for antiviral capabilities. During the early Th1-dominant immune response, early clinical symptoms arise, such as fever, cough, diarrhea, dyspnea, and fatigue [63,64].

Recent studies have found that allergic asthma, which is characterized by a Th2-skewed immune response, might not only fail to exacerbate COVID-19 but could possibly reduce the severity of symptoms in some cases. This may be due to the Th2 immune response predominating the Th1 responses [65], which are more associated with severe viral infections and inflammatory responses. Specifically, it has been found that the release of the Th2 cytokines IL-4 and IL-13 lead to the reduced activity of Th1 cells and the decreased production of pro-I inflammatory cytokines like IL-1 and IFN-γ [66]. This predominance of a Th2 immune response may provide a protective effect against severe SARS-CoV-2 symptoms in some cases of allergic asthma.

## 4. ACE2 and TMPRSS2 Expression

ACE-2 is most understood as converting angiotensin 1 into angiotensin 1–9 and angiotensin 2 into angiotensin 1–7. ACE-2 is expressed in many organ tissues but is found mainly in the nose, lung, heart, intestine, and kidney, with the greatest abundance on the surface of lung alveolar epithelial cells and enterocytes of the small intestine [17,67]. The characteristic S1 and S2 spike proteins of SARS-CoV-2 have been shown to bind to cell-surface ACE-2 enzymes to enter cells via endocytic mechanisms (Figure 3) [68,69,70]. The SARS-CoV-1 virus, which was responsible for a lesser pandemic in 2002, also binds to the ACE-2 receptor on cells, but SARS-CoV-2 is shown to bind with 10 times greater affinity [71]. Thus, the cellular expression of ACE-2 has become an increasing area of research in defining the disease pathology of SARS-CoV-2.

Research has shown that in those with atopic asthma, and asthma induced by allergens, there is a decreased ACE-2 expression [72]. Moreover, the expression of the enzyme appears to be in lowest concentration in the lower respiratory tract of asthmatics [73]. There is a small amount of confounding information found on the expression on ACE-2 in non-atopic asthmatic patients. Though, some studies have found that in those with non-atopic asthma diagnoses, the expression of ACE-2 is shown to be increased compared to non-affected airways [74].

In a cohort of COVID-19 infected patients in China, it was found that the number of ACE-2 positive airway cells was lower in those who had asthma than those who did not have the chronic respiratory condition [75]. The major sites of SAR-CoV-2 replication has been found to be nasopharynx and oropharynx [76]. The reduced number of ACE-2 positive cells throughout the airway of asthma patients might lead to a lower risk of infection or less viral replication in the airway tissues, providing a protective mechanism against the virus.

Furthermore, evidence has identified that IL-13 and other Th2 cytokines are key drivers of allergic inflammation and the patho-physiological changes in asthma, but may also be linked to reduced ACE-2 expression in asthmatic airway cells and an increase in transmembrane serine protease 2 (TMPRSS2) [77,78]. TMPRSS2 is an endothelial cell-surface protein that is also involved in the endocytic cell uptake of SARS-CoV-2 [79,80]. TMPRSS2 is used for S protein priming before SARS-CoV-2 cellular entry by cleaving the S1/S2 and S2’ sites to allow for viral and cell-membrane fusion [79]. Research focusing on the expression of ACE-2 and TMPRSS2 in the presence of IL-13 was conducted by collecting airway epithelial cells from patients with asthma via bronchoscopy. Cells were then cultured in the lab and exposed to IL-13. Statistical models were used to examine the effects of IL-13 on gene expression, showing that IL-13 reduces ACE2 and increases TMPRSS2 levels [78]. The dueling nature of increased TMPRSS2 but decreased ACE-2 expression may explain why asthma has not consistently been identified as a risk factor for severe COVID-19. TMPRSS2 plays a vital role in the cellular entry of SARS-CoV-2, and increased levels of the protein may allow for more enhanced viral entry. However, the reduction ACE2 may negate this factor and even offer additional protection against SARS-CoV-2.

Overall, emerging research is showing that asthma is not a risk factor for a poorer COVID-19 prognosis, and this seems to be bolstered by the body of scientific works showing differential ACE-2 expression in asthmatic airways [81].

### Inhaled Corticosteroid Treatment

ICSs are a first-line treatment for those suffering from chronic asthma. Two common medications taken for chronic asthmatic conditions are budesonide and fluticasone. The effects of these drugs can generally be seen within 24 h, with more pronounced systemic effects occurring after one to two weeks of consistent administration. Corticosteroids function by decreasing generalized airway inflammation, decreasing mucous production, and preventing destructive cellular functioning that can cause feedforward asthmatic responses. ICSs can provide measurable airway-saving effects with high airway responsiveness within six hours of dosage and can be crucial for airway rescue during asthmatic responses [82]. The predominate function that ICSs serve in asthmatic treatment is the downregulation of pro-inflammatory proteins and cellular functions that contribute to airway restriction [83,84,85,86].

On a genetic level, knowledge on the transcriptional regulatory functions that ICSs control is rapidly expanding [86]. ICSs have been shown to upregulate and downregulate the transcription of certain genes the encode proteins responsible for airway constriction, with the ultimate net effect of decreasing airway constriction [87,88]. ICSs also function by decreasing the amounts of pro-inflammatory cells in the airways such as eosinophils, T cells, mast cells, and dendritic cells [86].

Presumably, corticosteroids can decrease the pro-inflammatory effects that SARS-CoV-2 can have both systemically and on the airway. Patients that use regimented ICS treatments for chronic asthmatic conditions are shown to be affected on a systemic level by the ICSs [89]. SARS-CoV-2 increases the levels of pro-inflammatory mast cells, which can majorly contribute to cytokine storm inflammatory pathways [89].

SARS-CoV-2 upregulates many pro-inflammatory cytokines including IL-1, IL-2, IL-6, IL-7, IL-8, IL-12, IL-17, tumor necrosis factor-alpha (TNF-α), and GM-CSF (Figure 4) [89]. Further SARS-CoV-2 upregulates pro-inflammatory chemokines such as CCL2, CXCL10, and CXCL9 [90]. The upregulation of these cytokines and chemokines can be used to assess disease severity [75]. Together, the upregulation of the cytokines and chemokines contributes to the inflammatory response witnessed in many SARS-CoV-2 patients and contributes to the cytokine storm commonly seen in patients in phase three of viral infection [89,91]. Corticosteroids have been shown to directly downregulate the production of some of the pro-inflammatory cytokines such as IL-1, IL-2, IL-6, and TNF-α, and are shown to directly inhibit the cytokine storm associated with SARS-CoV-2 (Figure 5) [90,91,92].

The complex management of SARS-CoV-2 has been at the forefront of the research and clinical management in recent years. Recently, systemic corticosteroid treatment has emerged as a key therapeutic option for patients infected with SARS-CoV-2, decreasing hospital mortality rates and improving the clinical outcomes of patients, particularly those requiring supplemental oxygen [89,91]. The benefit of corticosteroids may be especially pronounced among critically ill patients, as they modulate the hyperinflammatory response associated with severe COVID-19.

Further, it has been shown that the earlier administration of corticosteroids to patients with SARS-CoV-2 is associated with better outcomes [93]. A study conducted on the timing of corticosteroid use in patients with severe COVID-19 demonstrated that those who received corticosteroids within the first 10 days of diagnosis experienced shorter hospital stays and faster improvement in clinical markers like c-reactive protein (CRP), lymphocyte count, and PaO_2_/FiO_2_ ratio, compared to those who received them later. Patients in the early-use group had a mean hospital discharge time of 27.8 days, while the late-use group were discharged at 57.4 days. However, both groups showed improvements in inflammation markers [94]. Another study found that initiating corticosteroid use with the first 48 h of hospitalization lead to a shorter length of hospital stays, as well as lower rates of care escalations to intensive care units, mechanical ventilation, and in-hospital mortality [95]. Results like these suggest that early use of corticosteroids may mitigate the inflammatory nature of COVID-19.

ICSs and corticosteroids, in general, are an essential mainstay in the management of chronic asthma, providing improvement in airway function and reduction in exacerbations. The potential for ICSs to mitigate airway inflammation and cytokine stores in patients with COVID-19, warrants further research. However, the early administration of systemic corticosteroids has shown considerable efficacy in treating severe COVID-19, improving clinical outcomes.

## 5. Vaccination and Biologic Use

With widespread vaccination efforts during the COVID-19 pandemic, there are many potential interactions with asthma through direct immune system modulation or concurrent biologic use. Caminati et al. studied the effects of COVID-19 vaccination on severely asthmatic patients and found that in patients taking asthma biologics, less than 20% reported side effects. Furthermore, patients not taking biologics reported even fewer adverse effects [96]. There is little data on the interaction between COVID-19 antibody production and biologic use. However, some early data suggest that biological therapy does not significantly change therapeutic outcomes [97].

While the majority of data demonstrates that COVID-19 vaccination has positive outcomes for individuals with asthma, there are reports of asthma exacerbation due to vaccine administration. In a case report by Ando et al., the Pfizer BNT162b2 messenger RNA (mRNA) COVID-19 vaccine led to an asthma exacerbation in a 55-year-old female patient [98]. Uzer and Cilli published a case report where a 76-year-old female experienced an acute asthma exacerbation 1 day after receiving the CoronaVac whole inactivated COVID-19 vaccine [99]. Colaneri et al. also published a case report where a 28-year-old female had a severe asthma exacerbation 3 weeks after receiving a second dose of the Pfizer mRNA COVID-19 vaccine [100].

## 6. Conclusions

Although asthma is typically associated with an increased susceptibility to viral respiratory infections, acute exacerbations, as well as accelerated and aggressive disease progression, the COVID-19 pandemic presented an unexpected trend in asthma patients. Contrary to initial concerns, studies revealed that individuals with asthma did not experience poorer outcomes from SARS-CoV-2 infection.

Potential protective factors such as eosinophilia, Th2-skewed immune response, reduced ACE-2 expression, and the use of inhaled corticosteroids, may have contributed to the more favorable outcomes seen in asthmatic patients with COVID-19. Eosinophilia, a common asthmatic clinical feature, may enhance immune defense by degrading viral RNA and promoting antigen presentation and T cell proliferation, reducing viral load. Reduced ACE-2 expression in the airways of asthma patients may limit viral entry into cells, and the Th2 immune response, prominent in allergic diseases like atopic asthma, could suppress the pro-inflammatory Th1 response via the type 2 cytokines IL-4 and IL-13, offering further protection. Inhaled corticosteroids, commonly used in asthma management, the inflammatory response associated with cytokine storms, commonly seen in severe COVID-19 cases.

These findings highlight the complexity of immune responses in the context of SARS-CoV-2 in asthmatic patients. Further research is needed to better understand the mechanisms behind these protective factors and how they might inform future treatment strategies for both asthma and COVID-19.

## Figures and Tables

**Figure 1 ijms-26-01678-f001:**
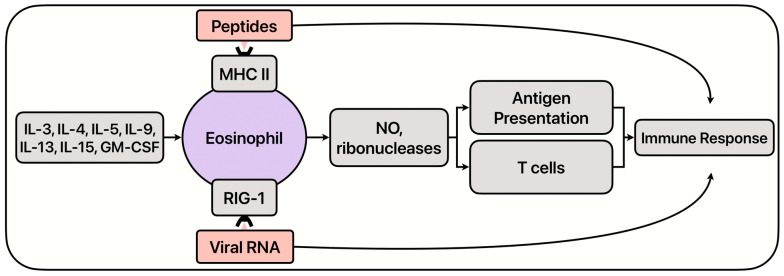
Basic overview of eosinophil mechanisms.

**Figure 2 ijms-26-01678-f002:**
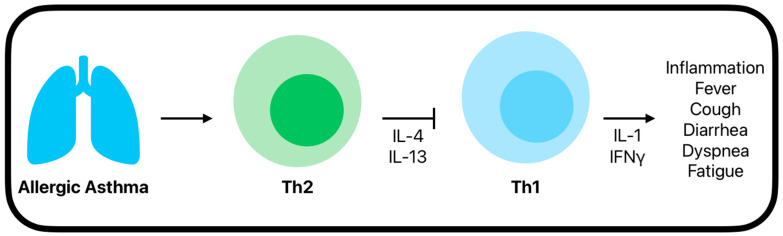
Th2 immune response in allergic asthma and its impact on Th1-mediated inflammation.

**Figure 3 ijms-26-01678-f003:**
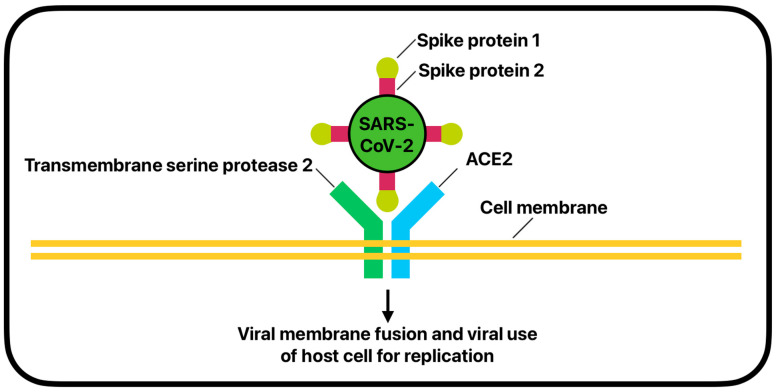
Angiotensin-converting enzyme 2 and transmembrane serine protease 2-mediated cellular entry by SARS-CoV-2.

**Figure 4 ijms-26-01678-f004:**
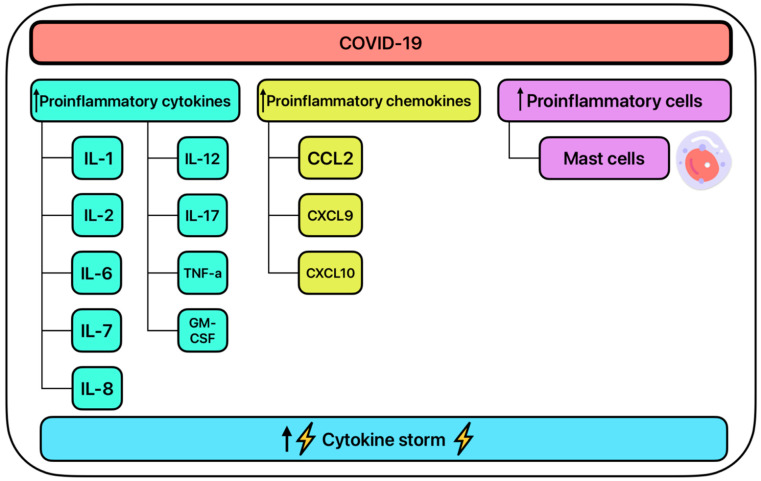
The systemic inflammatory effects of COVID-19.

**Figure 5 ijms-26-01678-f005:**
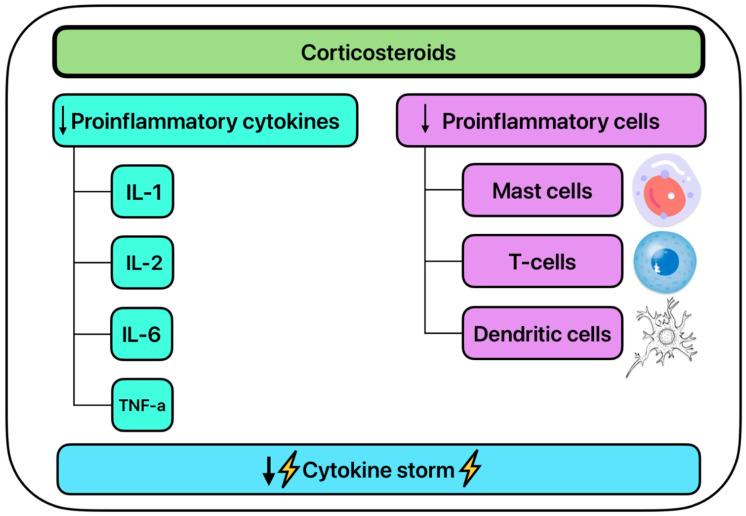
The systemic inflammatory effects of corticosteroids.

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
