# Peer review of "Exploring Asthma as a Protective Factor in COVID-19 Outcomes"

_ijms, 2025, doi:10.3390/ijms26041678_

Round 1

Reviewer 1 Report

Comments and Suggestions for Authors

The work of Quinn et al. is a mini review that Explores asthma as a protective factor in COVID-19 outcomes. The Ms is clearly written, concise with appropriate considerations on the possible protective mechanisms that asthmatic patients might display that make them more resistant to COVID-19. The work might be of interest for the broad readership of Int. J. Mol. Sci.

Reviewer 2 Report

Comments and Suggestions for Authors

The review is based on a current topic, using new research findings. The review analyzes the impact of asthma on patients infected with SARS-CoV-2 and the risk of severe infection. It provides data on the overall protective effect against SARS-CoV-2 infection through various mechanisms, including eosinophilia, inhaled steroid use, reduced ACE expression, and the dominance of type 2 inflammation in asthma.

I missed several important questions that would be of interest to readers: the effect of SARS-CoV-2 vaccines and biologics on the severity of COVID-2019 in asthma. It would be desirable to supplement the review in these aspects.
